# Nutritional Status and Sleep Quality Are Associated with Atrial Fibrillation in Patients with Obstructive Sleep Apnea: Results from Tokyo Sleep Heart Study

**DOI:** 10.3390/nu15183943

**Published:** 2023-09-12

**Authors:** Kazuki Shiina, Yoshifumi Takata, Takamichi Takahashi, Junya Kani, Hiroki Nakano, Yasuyuki Takada, Yoshinao Yazaki, Kazuhiro Satomi, Hirofumi Tomiyama

**Affiliations:** Department of Cardiology, Tokyo Medical University, 6-7-1 Nishishinjuku, Shinjuku-ku, Tokyo 160-0023, Japan; takatamd@gb3.so-net.ne.jp (Y.T.); taka.taka.0463@gmail.com (T.T.); junya.kani@gmail.com (J.K.); hiroki08192001@gmail.com (H.N.); takadaya@tokyo-med.ac.jp (Y.T.); yazaki@tokyo-med.ac.jp (Y.Y.); ksatomi@tokyo-med.ac.jp (K.S.); tomiyama@tokyo-med.ac.jp (H.T.)

**Keywords:** obstructive sleep apnea, atrial fibrillation, nutrition, sleep quality, Japanese

## Abstract

The prevalence of obstructive sleep apnea (OSA) in patients with atrial fibrillation (AF) has been observed to be much higher than in control participants without AF. Limited data exist regarding the prevalence of AF in patients with OSA. The clinical characteristics, nutritional status, and sleep parameters associated with AF in patients with OSA remain unclear. In this study, we aimed to determine the prevalence and factors associated with AF in patients with OSA from a large Japanese sleep cohort (Tokyo Sleep Heart Study). This was a single-center explorative cross-sectional study. Between November 2004 and June 2018, we consecutively recruited 2569 patients with OSA who underwent an overnight full polysomnography at our hospital. They were assessed using a 12-lead ECG and echocardiography. The clinical characteristics, sleep parameters, and medical history were also determined. Of the OSA patients, 169 (6.6%) had AF. Compared with the non-AF patients, OSA patients with AF were older and male, and they had higher prevalence of a history of alcohol consumption, hypertension, chronic kidney disease, and undernutrition, as well as a reduced ejection fraction. With regard to the sleep study parameters, OSA patients with AF had reduced slow-wave sleep and sleep efficiency, as well as higher periodic limb movements. There were no significant differences in the apnea–hypopnea index or hypoxia index between the two groups. The logistic regression analysis demonstrated that age (OR = 4.020; 95% CI: 1.895–8.527; *p* < 0.001), a history of alcohol consumption (OR = 2.718; 95% CI: 1.461–5.057; *p* = 0.002), a high CONUT score (OR = 2.129; 95% CI: 1.077–4.209; *p* = 0.030), and reduced slow-wave sleep (OR = 5.361; 95% CI: 1.505–19.104; *p* = 0.010) were factors significantly related to AF. The prevalence of AF in patients with OSA was 6.6%. Age, a history of alcohol consumption, undernutrition, and reduced sleep quality were independent risk factors for the presence of AF in patients with OSA, regardless of the severity of OSA.

## 1. Introduction

Obstructive sleep apnea (OSA) is identified as one of the risk factors contributing to the onset of atrial fibrillation (AF) [1,2,3]. It plays a role in promoting arrhythmogenesis [4,5] and hindering the effectiveness of treatments [6,7,8,9]. Previous studies have shown that the occurrence of OSA in patients with AF is considerably higher (ranging from 21% to 74%) compared to those without AF [1,10,11,12,13]. Conversely, the occurrence of AF in patients with OSA has been estimated to be between 3% and 6% [14,15,16], which is somewhat higher than in control patients or the general population. However, there is limited available data on the prevalence of AF in patients with OSA.

With OSA, the combination of high-frequency intermittent hypoxia, fluctuations in intrathoracic pressure, stretching of the atrium, and increased activation of sympathetic nerves due to arousals leads to a gradual atrial structural remodeling [10,17]. This ongoing process of atrial structural remodeling, coupled with electrophysiological alterations linked to intermittent apneas, establishes a multifaceted and evolving arrhythmogenic foundation for AF. The specific influence of each element of OSA, such as intermittent hypoxia, intrathoracic pressure changes, and sympathetic activation, is not yet fully understood.

Obesity, a major risk factor for OSA [18], has been reported to contribute to AF through sympathetic nerve activation [19,20,21]. Recent studies have also suggested that undernutrition may play a role in the development of AF [19,22,23]. Wu et al. suggested that undernutrition was an independent predictor of new-onset AF in acute myocardial infarction patients [23]. However, there is no reported association between nutritional status, such as obesity, overweight, excessive nutritional status and undernutrition, and AF in patients with OSA.

In this study, we aimed to determine the prevalence of AF and discover the clinical factors predictive of AF in patients with OSA from a large Japanese sleep cohort (Tokyo Sleep Heart Study) [24]. In particular, we examined the associations between nutritional status, sleep quality, and the presence of AF in patients with OSA.

## 2. Methods

The Tokyo Sleep Heart Study is an ongoing large-scale cohort study conducted at the Tokyo Medical University Sleep Laboratory, which began enrolling Japanese patients in November 2004. This study aims to investigate the influence of sleep disorder breathing on cardiovascular disease [24].

### 2.1. Patient Selection

Between November 2004 and June 2018, a total of 3370 patients who had undergone polysomnography at Tokyo Medical University were consecutively enrolled in this study. Figure 1 illustrates the selection process of the study cohort. The inclusion criteria comprised: (1) age ≥ 30 years; (2) diagnosis of obstructive sleep apnea (defined as an apnea-hypopnea index (AHI) ≥ 5/h) based on an overnight polysomnography; (3) evaluation using a 12-lead ECG or examination of medical records; and (4) assessment via echocardiography. Subsequently, blood samples were collected for biochemical analysis and routine blood examinations. A total of 2569 patients were included in the analyses. Hypertension was defined as a systolic/diastolic blood pressure of ≥140/90 mmHg and/or current use of antihypertensive medications. Diabetes mellitus was determined by a prior diagnosis in medical records, a hemoglobin A1c value (as calculated by the National Glycohemoglobin Standardization Program) of ≥6.5%, or treatment with oral antidiabetic medications or insulin. Chronic kidney disease (CKD) was characterized by an estimated glomerular filtration rate (eGFR) < 60 mL/min/1.73 m^2^. The study was conducted with the approval of the Ethical Guidelines Committee of Tokyo Medical University (No. T2019-0216) in adherence to the principles of the Declaration of Helsinki.

The inclusion criteria were as follows: (1) age ≥ 30 years; (2) diagnosis of OSA (AHI ≥ 5/h), based on overnight polysomnography; (3) assessment using a 12-lead ECG; and (4) assessment using echocardiography. The blood was then collected for blood biochemistry and routine blood examinations (5) and participants completed a written informed consent form (6). Finally, a total of 2569 patients were included in this study. OSA: obstructive sleep apnea; PSG: polysomnography; AHI: apnea–hypopnea index; ECG: electrocardiogram; AF: atrial fibrillation.

### 2.2. ECG Examinations and Definition of AF

For all patients included in the study who underwent a 12-lead ECG assessment, the clinical diagnosis of AF was established based on the following criteria. The primary electrocardiographic indicators of AF included the absence of P waves, replaced by fibrillatory waves; irregular and rapid activation of the ventricles, leading to an irregular heart rate; and a narrow QRS complex, unless there were concurrent conduction abnormalities. Paroxysmal AF was characterized as an episode of AF that spontaneously resolved, or following the administration of antiarrhythmic drugs within 7 days of onset. Persistent AF was defined as an AF episode lasting for ≥7 days and up to 1 year. Long-standing AF was classified as persisting for ≥1 year.

### 2.3. Assessment of Nutritional Status 

We assessed nutritional status at admission for polysomnography using the CONUT scoring system to estimate the values of the nutrition-related prognostic risk factors. The CONUT scores were calculated from total peripheral lymphocyte counts, the serum albumin levels, and total cholesterol levels, as described previously [25]. In this study, CONUT scores of 0–1 point were defined as normal CONUT scores and CONUT scores ≥ 2 points were defined as high CONUT scores [26]. Using the Nutritional Assessment as the gold standard approach, the CONUT score had a sensitivity of 92.3 and a specificity of 85.0 [25].

### 2.4. Sleep Study 

Fully attended polysomnography (PSG) was performed overnight at the Tokyo Medical University Sleep Laboratory using an Alice 6 Sleep System TM (Respironics, Inc., Murrysville, PA, USA). Apnea-hypopnea episodes, sleep stages, and arousal were scored according to standard criteria [27]. The AHI was calculated as the total number of apnea–hypopnea events per hour of sleep. Patients were diagnosed with OSA when an obstructive component accounted for more than 50% of the apnea–hypopnea events and the AHI was 5 per hour. The severity of OSA was classified according to the criteria of the American Academy of Sleep Medicine [27].

### 2.5. Statistical Analysis 

The continuous variables were expressed as the mean ± standard deviation or the median with an interquartile range. The categorical variables were presented as numbers and percentages. Parametric data were compared using the Student’s *t*-test. Non-parametric data were compared using the Mann–Whitney U test, Wilcoxon signed-rank test, chi-squared test, or Fisher’s exact test as appropriate. A multivariable logistic regression analysis was performed to determine the risk factors for AF using the following variables: age ≥ 65 years old, male sex, a history of alcohol consumption, hypertension, CKD, a CONUT score ≥ 2, slow-wave sleep < 10%, sleep efficiency < 75%, and PLM ≥ 5/h. The odds ratios (95% confidence intervals) and *p*-values were calculated. Two-sided *p*-values < 0.05 were considered to denote statistical significance. Data management and statistical analyses were performed using SPSS software (IBM), version 28.

## 3. Results

### 3.1. Baseline Characteristics

The characteristics of the patients are shown in Table 1. A total of 2569 patients were included in this study. In the overall patient population, the mean age was 56 ±13 years, the mean BMI was 26.8 ± 4.7 kg/m^2^, and 85% of the patients were men. Compared with the non-AF patients, OSA patients with AF were older, predominantly male, and had higher prevalence of a history of alcohol consumption, hypertension, CKD, and a higher CONUT score as well as a lower ejection fraction.

### 3.2. The Prevalence of AF in Patients with OSA (Figure 2)

The prevalence of AF was 6.6% (*n* = 159) in patients with OSA (*n* = 2569). In the analysis, a higher frequency of AF was observed in cases with moderate (6.9%) to severe (7.2%) OSA than in those with mild OSA (3.9%) when based on the severity of OSA. When considering the classification of AF types, there was a higher incidence of paroxysmal AF (3%) and long-standing AF (2.6%).

**Figure 2 nutrients-15-03943-f002:**
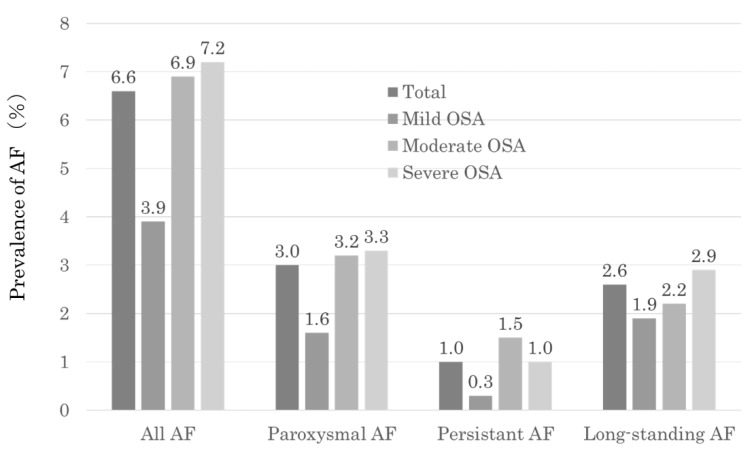
Prevalence of AF in the total study population and severity of AF classes.

### 3.3. Sleep Study

The sleep study findings are shown in Table 2. OSA patients with AF had a lower total sleep time, a lower percentage of slow-wave sleep, lower sleep efficiency, and a higher periodic limb movement (PLM) index. There were no significant differences in the AHI or hypoxia index (3% ODI, lowest SpO_2_, and sleep time with SpO_2_ < 90%) between the two groups.

### 3.4. Factors Associated with the Presence of AF in OSA Patients

Univariate and multivariate logistic regression analyses were performed to determine the risk factors of AF in patients with OSA (Table 3). The significant risk factors of AF were age, male sex, a history of alcohol consumption, hypertension, CKD, a CONUT score ≥ 2%, slow-wave sleep < 10%, sleep efficiency < 75%, and PLM index ≥ 5/h in the univariate analysis. The multivariate logistic regression analysis identified that age ≥ 65 (OR = 4.020; 95% CI: 1.895–8.527; *p* < 0.001), a history of alcohol consumption (OR = 2.718; 95% CI: 1.461–5.057; *p* = 0.002), a CONUT score ≥ 2 (OR = 2.129; 95% CI: 1.077–4.209; *p* = 0.030), and % of slow-wave sleep < 10% (OR = 5.361; 95% CI: 1.505–19.104; *p* = 0.010) were factors significantly related to the presence of AF.

## 4. Discussion

This is the first study to demonstrate that nutritional status and sleep quality were associated with the presence of AF in patients with OSA from a large Japanese sleep cohort (Tokyo Sleep Heart Study) [24]. In this study, all OSA patients were accurately diagnosed by performing a full PSG, and all OSA patients also underwent electrocardiography and echocardiography to accurately assess AF and cardiac functions. The major findings were as follows: (1) the prevalence of AF was 6.6% in patients with OSA. (2) The CONUT score was significantly associated with the presence of AF in OSA patients. (3) Reduced slow-wave sleep, which indicates poor sleep, was also significantly associated with the presence of AF. (4) Among the lifestyle habits, a history of alcohol consumption was significantly associated with the presence of AF.

### 4.1. The Prevalence of AF in Patients with OSA

In this study, among patients with OSA obtained from a large-scale Japanese sleep cohort (*n* = 2569), the prevalence of AF was 6.6% (*n* = 159). Previous studies have reported AF prevalence in OSA patients ranging from 3% to 6% [14,15,16]. Our analysis revealed that, based on the severity of OSA, there was a higher frequency of AF observed in cases with moderate to severe OSA. When classifying the types of AF, there was a higher incidence of paroxysmal AF and long-standing AF (Figure 2). The prevalence of AF varies depending on the characteristics of the study population and the diagnostic criteria used, making the exact prevalence uncertain. The risk of OSA inducing new-onset AF increases over time [2]; therefore, the prevalence of AF co-occurrence becomes higher as the duration of illness lengthens. Given the frequent coexistence of AF and OSA, it is crucial to assess the simultaneous risk of AF in patients with OSA and consider OSA screening when evaluating individuals with AF.

### 4.2. Nutritional Status and AF in OSA Patients

The precise role of nutritional status in the pathogenesis of AF remains unclear. In a systematic review [19], Anaszewicz et al. found conflicting opinions regarding how nutritional status affects the risk, progression, and complications of AF. Both being overweight and obesity have been associated with an increased risk of AF [19,20,21], whereas reducing these factors has been linked to a better course of AF [28]. Conversely, it has been reported that undernutrition is also a contributing factor to the onset of AF [19,22,23]. Recently, Furui et al. noted that the recurrence rate of AF after catheter ablation was higher in patients with undernutrition compared to those with normal nutrition [29]. Additionally, although based on a limited number of studies, it has been suggested that the relationship between nutritional status and the risk of AF and its complications may exhibit a U-shaped pattern [30]. As a result, individuals who are obese, underweight, or affected by cachexia may face an elevated risk of AF and suboptimal outcomes [31]. In this study, we did not find a significant correlation between body mass index and AF in patients with OSA. However, undernutrition, as assessed by the CONUT score, demonstrated a significant association with the presence of AF. There are two potential explanations for these results. First, our analysis was adjusted for other variables that held greater predictive power and controlled for confounding factors such as age, a history of alcohol consumption, the CONUT score, and sleep parameters. In comparison, body mass index did not emerge as a strong predictor of AF presence in OSA patients, relative to the other variables examined in this study. Second, this study was conducted using Japanese patients with OSA who, in comparison with Western populations, have lower levels of obesity [32]. The OSA patients with AF in this study may have exhibited a different phenotype from the severely obese AF patients with severe OSA commonly observed in Western populations.

### 4.3. Sleep Quality and AF in OSA Patients

In patients with OSA, intermittent hypoxia, sympathetic nervous system activation, fluctuations in intrathoracic pressure, atrial stretching, and the presence of chronic concurrent conditions like obesity, metabolic syndrome, and hypertension collectively contribute to a gradual atrial structural remodeling [10,17]. Along with transient apnea-associated electrophysiological changes, this progressive atrial structural remodeling contributes to a re-entry substrate for AF, creating a complex and dynamic arrhythmogenic substrate in the atrium during sleep. Among the components of OSA, the specific sleep indices that impact the presence of AF have not been thoroughly investigated. The results of this study indicate a novel perspective, highlighting that it is not traditional indices such as the AHI or oxygen desaturation index that are associated with the presence of AF, but rather the proportions of deep sleep, sleep efficiency, and PLM. It has been reported that the proportion of deep sleep decreases with age [33], suggesting a significant age-related impact. In this study, even after adjusting for age, a significant reduction in deep sleep remained associated with the presence of atrial AF. These results suggest that when considering the risk of AF complications in patients with OSA, it is important to not only assess the severity of OSA, but also pay attention to the indices related to sleep quality, such as the proportion of slow-wave sleep. While this is not necessarily a study on sleep duration, those with AF had lower total sleep time in this study, which may indicate that sleep duration and AF could be linked. Further studies are needed to explore the association between sleep time and AF.

### 4.4. Study Limitations

This study had several limitations. Firstly, it was a single-center cross-sectional study. However, we are confident that the findings derived from our clinical practice are comparable to those of a well-designed prospective study, as we conducted comprehensive overnight full PSG for OSA diagnosis and utilized both 12-lead ECG and electrocardiography for AF diagnosis. Additionally, we gathered detailed medical histories from all patients, and the sample size was substantial (*n* = 2569). Secondly, the study’s participant pool was exclusively composed of Japanese individuals, potentially introducing selection bias, such as a higher proportion of non-obese patients. Finally, it is possible that the CONUT score may have been influenced by low plasma cholesterol levels resulting from statin therapy. Nonetheless, we observed no significant associations between lipid-lowering medication and the incidence of AF (Table 3).

## 5. Conclusions

In this large Japanese sleep cohort of 2569 OSA patients who underwent complete PSG, the prevalence of AF was 6.6%. Undernutrition and reduced sleep quality were independent risk factors for the presence of AF. As the risk of new-onset AF induced by OSA increases over time, the identification of potentially modifiable risk factors in patients with OSA, such as undernutrition and poor quality of sleep, may have significant implications for the prevention of AF.

## Figures and Tables

**Figure 1 nutrients-15-03943-f001:**
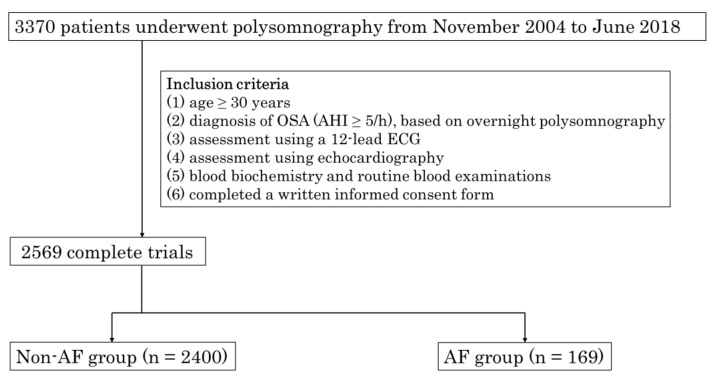
Flowchart of selection of the study population. From November 2004 to June 2018, 3370 patients consecutively underwent polysomnography at the Tokyo Medical University.

**Table 1 nutrients-15-03943-t001:** Clinical characteristics of the study subjects.

Variables	All (*n* = 2569)	Non-AF Group (*n* = 2400)	AF Group (*n* = 169)	*p*-Value
Age (y)	56 ± 13	56 ± 13	64 ± 9	<0.001
Male, *n* (%)	2173 (85)	2016 (84)	157 (93)	<0.001
BMI (kg/m^2^)	26.8 ± 4.7	26.8 ± 4.7	26.4 ± 3.9	0.306
History of smoking, *n* (%)	628 (25)	576 (24)	52 (31)	0.051
History of alcohol consumption, *n* (%)	1259 (49)	1152 (48)	107 (63)	<0.001
SBP (mmHg)	127 ± 16	128 ± 16	125 ± 17	0.017
DBP (mmHg)	76 ± 11	76 ± 11	74 ± 11	0.004
Heart rate (per min)	72 ± 13	71 ± 12	73 ± 14	0.057
Total protein (mg/dL)	7.1 ± 0.5	7.1 ± 0.5	6.9 ± 0.5	<0.001
Albumin (g/dL)	4.2 ± 0.4	4.3 ± 0.4	4.0 ± 0.4	<0.001
Lymphocyte count (per mm^3^)	1974.0 ± 855.2	2110.5 ± 939.7	1778.6 ± 673.6	<0.001
T-cho (mg/dL)	198 ± 37	199 ± 37	189 ± 33	<0.001
CONUT score	0.9 ± 1.3	0.6 ± 1.0	1.4 ± 1.6	<0.001
HDL-cho (mg/dL)	48 ± 13	48 ± 14	47 ± 11	0.411
LDL-cho (mg/dL)	116 ± 30	117 ± 31	109 ± 26	0.003
Triglyceride (mg/dL)	162 ± 94	162 ± 96	157 ± 94	0.467
Fasting plasma glucose (mg/dL)	98 ± 28	98 ± 27	101 ± 29	0.196
HbA1c (%)	6.1 ± 1.0	6.1 ± 1.0	6.2 ± 0.9	0.299
Uric acid (mg/dL)	6.2 ± 2.4	6.2 ± 2.4	6.3 ± 1.4	0.654
Cr (mg/dL)	0.95 ± 1.08	0.94 ± 1.04	1.14 ± 1.50	0.041
eGFR (mL/min/1.73 m^2^)	107.7 ± 41.8	109.1 ± 42.0	88.5 ± 31.9	<0.001
Hypertension, *n* (%)	1514 (59)	1392 (58)	122 (72)	<0.001
Dyslipidemia, *n* (%)	1740 (68)	1632 (68)	108 (64)	0.215
Diabetes mellitus, *n* (%)	572 (22)	528 (22)	44 (26)	0.205
CKD, *n* (%)	269 (11)	240 (10)	29 (17)	0.018
Coronary heart disease, *n* (%)	963 (37)	912 (38)	51 (30)	0.627
Ejection fraction (%)	65 ± 6	66 ± 6	62 ± 6	<0.001
Medication				
Antihypertensive, *n* (%)	1297 (50)	1152 (48)	145 (86)	<0.001
Antidiabetic, *n* (%)	284 (11)	264 (11)	20 (12)	0.629
Lipid-lowering, *n* (%)	709 (28)	648 (27)	61 (36)	0.013

Values are mean ± SD or No. (%). *p*-values are between non-AF group and AF group. BMI: body mass index; SBP: systolic blood pressure; DBP: diastolic blood pressure; T-cho: total cholesterol; CONUT: controlling nutritional status; HDL-cho: serum high-density lipoprotein cholesterol; LDL-cho: serum low-density lipoprotein cholesterol; Cr: serum creatinine; eGFR: estimated glomerular filtration rate; CKD: chronic kidney disease.

**Table 2 nutrients-15-03943-t002:** Sleep-study findings.

Variables	All (*n* = 2569)	Non-AF Group (*n* = 2400)	AF Group (*n* = 169)	*p*-Value
Total sleep time (min)	423 ± 64	425 ± 63	397 ± 66	<0.001
% of slow-wave sleep, % of TST	5.0 ± 5.6	5.1 ± 5.6	3.0 ± 3.9	<0.001
% of REM sleep, % of TST	17.7 ± 6.2	17.8 ± 6.2	16.9 ± 6.2	0.092
Sleep efficiency (%)	80 ± 11	80 ± 10	76 ± 12	<0.001
Arousal index (event/h of sleep)	45 ± 20	45 ± 20	46 ± 18	0.641
AHI (event/h)	40.4 ± 22.7	40.4 ± 22.9	40.9 ± 19.6	0.772
3% ODI (event/h)	31.5 ± 22.7	31.6 ± 23.0	30.6 ± 18.6	0.587
Lowest SpO_2_ (%)	77.4 ± 12.0	77.3 ± 12.0	78.1 ± 11.9	0.405
Sleep time with SpO_2_ < 90% (%)	6.4 ± 11.8	6.5 ± 12.0	4.7 ±8.0	0.058
PLM index (event/h)	11.3 ± 23.7	10.6 ± 22.5	20.7 ± 35.8	<0.001

Values are mean ± SD or No. (%). *p*-values are between non-AF group and AF group. TST: total sleep time; REM: rapid eye movement; AHI: apnea–hypopnea index; ODI: oxygen desaturation index; PLM: periodic limb movement.

**Table 3 nutrients-15-03943-t003:** Factors associated with the presence of AF in patients with OSA.

Factors	Univariate		Multivariate	
	Odds Ratio (95% CI)	*p*-Value	Odds Ratio (95% CI)	*p*-Value
Age ≥ 65	2.924 (2.134–4.007)	<0.001	4.020 (1.895–8.527)	<0.001
Male	2.578 (1.419–4.682)	0.002	2.938 (0.946–9.127)	0.062
BMI ≥ 25 (kg/m^2^)	0.982 (0.949–1.017)	0.306		
History of smoking	1.401 (0.997–1.968)	0.052		
History of alcohol consumption	1.811 (1.302–2.519)	<0.001	2.718 (1.461–5.057)	0.002
Hypertension	1.801 (1.273–2.550)	<0.001	1.374 (0.927–2.035)	0.113
Dyslipidemia	0.813 (0.585–1.129)	0.216		
Diabetes mellitus	1.265 (0.879–1.822)	0.206		
CKD	1.943 (1.244–3.036)	0.004	1.016 (0.602–1.716)	0.952
Coronary heart disease	0.704 (0.173–2.872)	0.625		
Medication				
Antihypertensive	5.561 (3.657–8.456)	<0.001		
Lipid-lowering	1.007 (0.702–1.444)	0.971		
CONUT score ≥ 2	3.455 (2.101–5.681)	<0.001	2.129 (1.077–4.209)	0.030
Ejection fraction < 50%	1.377 (0.542–3.493)	0.501		
Sleep Study				
% of slow-wave sleep < 10%	3.272 (1.760–6.082)	<0.001	5.361 (1.505–19.104)	0.010
Sleep efficiency < 75%	1.892 (1.374–2.604)	<0.001	1.237 (0.848–1.803)	0.270
Arousal index ≥ 30/h	1.282 (0.870–1.890)	0.209		
AHI ≥ 30/h	1.254 (0.902–1.743)	0.178		
3% ODI ≥ 30/h	0.794 (0.700–1.314)	0.794		
Mean SpO_2_ < 90%	0.159 (0.022–1.152)	0.069		
Lowest SpO_2_ < 80%	0.750 (0.546–1.030)	0.076		
Sleep time with SpO_2_ < 90% ≥ 5%	0.719 (0.499–1.036)	0.077		
PLM index ≥ 5/h	1.664 (1.211–2.286)	0.002	1.968 (0.968–4.004)	0.062

See Table 1 and Table 2 legends for definitions of abbreviations.

## Data Availability

The data supporting the findings of this study are available from the corresponding author upon reasonable request.

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
