# Peer review of "Nutritional Status and Sleep Quality Are Associated with Atrial Fibrillation in Patients with Obstructive Sleep Apnea: Results from Tokyo Sleep Heart Study"

_nutrients, 2023, doi:10.3390/nu15183943_

Round 1
Reviewer 1 Report
The article is interesting and extremely relevant as AF in this population is quite understudied. I appreciate the authors for taking up the topic. The article is structured well, and the information provided is good.
There are only a few minor issues that should be addressed:
1. The article would benefit if the authors could read for grammar and sentence structure.
2. It would be helpful if the authors expanded the introduction to include more information on what they mean by "nutritional status" as it is unclear if the authors are referring to obesity and/or another concept
3. While this isn't necessarily a study on sleep duration, the finding that those with AF had lower TST is quite interesting, as it indicates that sleep duration and AF could be linked. It may be worth it to add a line to the future directions indicating that TST/sleep duration should be explored in future studies.
Please see point 1 in the review above. Thanks!
Author Response
Responses to Reviewers' comments 1:
Thank you for considering our manuscript. We have responded to the reviewers' comments. Please note that if you or the reviewers still have concerns about our article, we are willing to revise it further.
Here are additional comments for your consideration:
The article is interesting and extremely relevant as AF in this population is quite understudied. I appreciate the authors for taking up the topic. The article is structured well, and the information provided is good.
There are only a few minor issues that should be addressed:
- The article would benefit if the authors could read for grammar and sentence structure.
Our manuscript submitted to MDPI for English editing has been edited.
- It would be helpful if the authors expanded the introduction to include more information on what they mean by "nutritional status" as it is unclear if the authors are referring to obesity and/or another concept
We would like to thank the reviewer for his/her important comments. We revised this sentence below.
“In the Japanese study [23], in addition to confirming the existence of the obesity paradox in AF, it indicated that a low body mass index (BMI) was associated with a significantly higher risk of all-cause death and a greater risk of composite endpoints, defined as all cause death, acute coronary syndrome, stroke, or transient ischemic attack, or heart failure requiring hospitalization. Currently, there is no reported association between nutritional status, such as obesity, overweight, excessive nutritional status and under-nutrition, and AF in patients with OSA.”
- While this isn't necessarily a study on sleep duration, the finding that those with AF had lower TST is quite interesting, as it indicates that sleep duration and AF could be linked. It may be worth it to add a line to the future directions indicating that TST/sleep duration should be explored in future studies.
We would like to thank the reviewer for his/her important comments. We revised this sentence below.
“While this isn't necessarily a study on sleep duration, those with AF had lower total sleep time in this study, as it may indicate that sleep duration and AF could be linked. Further studies are needed to explore the association between sleep time and AF.”
Reviewer 2 Report
The introduction is concise and to the point.
Methods
"The Tokyo Sleep Heart Study is a single center, large sleep cohort study of Japanese 61 with sleep apnea, enrolled at Tokyo Medical University Hospital Sleep Laboratory, from November 2004" ... Add termination date, please.
2.2. it is not possible "For all patients enrolled in the study who were assessed using12-lead ECG, the clin- 92 ical diagnosis of AF was based on the ACC/AHA/ESC 2016" when we have patients from 2004 ....
The results are presented in a correct manner.
The discussion is interesting and supported by current research. The only thing missing is a citation in line 213 "(...) an increased risk and poor outcome of AF". This study shows that undernutrition and obesity had impact for outcomes in AF patients doi.org/10.3389/fnut.2022.1086715
The conclusions are supported by the results
minior
Author Response
Thank you for considering our manuscript. We have responded to the reviewers' comments. Please note that if you or the reviewers still have concerns about our article, we are willing to revise it further.
Here are additional comments for your consideration:
The introduction is concise and to the point.
Methods
"The Tokyo Sleep Heart Study is a single center, large sleep cohort study of Japanese 61 with sleep apnea, enrolled at Tokyo Medical University Hospital Sleep Laboratory, from November 2004" ... Add termination date, please.
→We would like to thank the reviewer for his/her important comments. This study is currently ongoing, and we have made a note of this.
“The Tokyo Sleep Heart Study is a single-center, large sleep cohort study of Japanese patients with sleep apnea who were enrolled at the Tokyo Medical University Hospital Sleep Laboratory from November 2004 and is ongoing.“
2.2. it is not possible "For all patients enrolled in the study who were assessed using12-lead ECG, the clin- 92 ical diagnosis of AF was based on the ACC/AHA/ESC 2016" when we have patients from 2004 ....
→We would like to thank the reviewer for his/her important comments. We revised this sentence and delited ref.25.
“For all patients enrolled in the study who were assessed using a 12-lead ECG, the clinical diagnosis of AF was based on the following criteria.”
The results are presented in a correct manner.
The discussion is interesting and supported by current research. The only thing missing is a citation in line 213 "(...) an increased risk and poor outcome of AF". This study shows that undernutrition and obesity had impact for outcomes in AF patients doi.org/10.3389/fnut.2022.1086715
→We would like to thank the reviewer for his/her important comments. We added this new reference 30. Czapla M, Uchmanowicz I, Juárez-Vela R, Durante A, Kałużna-Oleksy M, Łokieć K, Baeza-Trinidad R, Smereka J. Relationship between nutritional status and length of hospital stay among patients with atrial fibrillation - a result of the nutritional status heart study. Front Nutr. 2022;9:1086715.
The conclusions are supported by the results